# Assays for Monitoring Apixaban and Rivaroxaban in Emergency Settings, State-of-the-Art Routine Analysis, and Volumetric Absorptive Microsamples Deliver Discordant Results

**DOI:** 10.3390/diagnostics14171939

**Published:** 2024-09-02

**Authors:** Adrienne Fehér, István Vincze, James Rudge, Gyula Domján, Barna Vásárhelyi, Gellért Balázs Karvaly

**Affiliations:** 1Department of Laboratory Medicine, Semmelweis University, H-1089 Budapest, Hungary; feher.adrienne@semmelweis.hu (A.F.); vincze.istvan@semmelweis.hu (I.V.); vasarhelyi.barna@semmelweis.hu (B.V.); 2Trajan Scientific & Medical, Milton Keynes MK8 0AB, UK; jrudge@trajanscimed.com; 3Department of Internal Medicine and Oncology, Semmelweis University, H-1083 Budapest, Hungary; domjan.gyula@semmelweis.hu

**Keywords:** direct acting oral anticoagulant, apixaban, rivaroxaban, volumetric absorptive microsampling, therapeutic drug monitoring, plasma, liquid chromatography, mass spectrometry, anti-Xa chromogenic assay

## Abstract

Our aim was to compare the performance of complementary clinical laboratory approaches to monitoring exposure to apixaban and rivaroxaban, the most prescribed direct-acting oral anticoagulants (DOAC’s): an automated commercial anti-Xa chromogenic assay suitable for emergency and pre-surgery testing and a laboratory-developed liquid chromatography-tandem mass spectrometry (LC-MS/MS) method employed for non-emergency analysis in plasma and in dried blood volumetric absorptive microsamples (VAMS) collectible by the patients in their homes. The full validation of the LC-MS/MS method was performed. Cross-validation of the methodologies was accomplished by processing 60 specimens collected for whole blood count and DOAC monitoring in a central clinical laboratory. For VAMS samples, dried plasma and whole blood calibrators were found to be suitable, and a cycle run for seven days could be implemented for rational and economic sample processing. The anti-Xa chromogrenic assay and the LC-MS/MS method delivered discordant plasma analyte concentrations. Moreover, the lack of agreement between plasma and VAMS concentrations was observed. Clinical laboratories must be aware of the differences between the performance of apixaban and rivaroxaban LC-MS/MS and anti-Xa assays. Hematocrit must always be measured along with VAMS samples to obtain accurate results.

## 1. Introduction

Oral anticoagulant therapy has changed since the introduction of direct-acting oral anticoagulants (DOACs). Today, apixaban (API) and rivaroxaban (RIV) account for about 75% of related prescriptions worldwide [1,2]. Overdose can lead to major or continuous bleeding, while inadequate dosing increases the risk of thromboembolic events [3,4,5]. A one-size-fits-all dosing concept is not appropriate due to the heterogeneity of the patient population, mostly comprising individuals who are elderly, suffer from comorbidities, and take multiple medications [6,7,8,9]. Monitoring API and RIV plasma concentrations is therefore particularly important in emergency and pre-surgery scenarios and is often useful in non-emergency settings [10].

Commercial functional assays and liquid chromatography-tandem mass spectrometry (LC-MS/MS) are concurrent technologies employed for monitoring API and RIV concentrations. Functional assays are based on the detection of a chromogenic substance cleaved from antithrombin complexed with heparin by the activated coagulation factor X (Xa), which is in turn inhibited in a concentration- and time-dependent manner by API and RIV. Anti-Xa chromogenic assays can be applied in emergency and pre-surgery care, but they are non-selective: the signal changes in response to any kind of inhibition of factor X. Consequently, combined oral antithrombotic therapy or an incorrect test order will lead to erroneous results [11].

For nonurgent testing, LC-MS/MS, which allows the direct quantitation of API and RIV with a high degree of specificity and selectivity, is considered to be the gold standard [12]. With purposeful method development, the quantitation of fingerprick capillary blood samples may also be feasible. Volumetric absorptive microsampling based on VAMS^®^ technology is an established patient-centric, convenient home-sampling methodology allowing the collection and drying of capillary blood samples by the patients themselves [13]. The application of this technology to monitoring API and RIV concentrations has not been investigated earlier. It relies on the application of a disposable, lightweight, non-electronic, bar-coded, plastic medical device: a small, yet precisely defined volume (10–30 µL) of blood is adsorbed onto a sterile porous polymer, and the device is allowed to sit in a packaging containing a desiccant or on a bench until the sample is dried completely (this takes approximately 3 h). While pricking one’s finger may obviously cause slight and temporary inconvenience, the collection of the VAMS sample is entirely painless and safe. The device containing the dried blood sample can be transported or, by using the packaging provided by its manufacturer, mailed to the testing laboratory without any special precautions.

Analyte recovery from dried blood depends, in addition to factors related to the chemical properties of the substance and the methodology employed, on the sample hematocrit. In this paper, we show that API and RIV recoveries are also related to their concentrations and propose an integrated analytical approach as well as mathematical equations for correcting the impact of these factors.

In the present work, a clinical LC-MS/MS method applicable to API and RIV measurements using plasma and dried whole blood volumetric absorptive microsamples (VAMS) is described. To increase the clinical feasibility and cost-efficiency of collecting VAMS samples for assaying API and RIV, various approaches to calibration are compared, and a sample processing cycle lasting seven days is evaluated. We also compare API and RIV levels measured in plasma and VAMS samples, as well as those obtained in the same blood specimens by performing the commercial anti-Xa chromogenic assay suitable for emergency testing.

## 2. Materials and Methods

Trajan Neoteryx^®^ Mitra^®^ VAMS^®^ 20-µL microsampling devices were bought from Buda Labor Kft. (Budapest, Hungary). API, ^13^C,^2^H_8_-API, RIV, and ^13^C_6_-RIV were purchased from Alsachim S.A.S. (Illkirch, France). LC-MS-grade acetonitrile, formic acid, methanol, and water were obtained from Reanal Labor Kft. (Budapest, Hungary). The internal standard (IS) solution contained ^13^C,^2^H_8_-API and ^13^C_6_-RIV at a concentration of 140 ng/mL in acetonitrile. In the experiments performed with VAMS samples, the extraction solution contained 35.0 ng/mL and 16.7 ng/mL of ^13^C,^2^H_8_-API and ^13^C_6_-RIV, respectively.

LC-MS/MS assays were performed at the Laboratory of Mass Spectrometry and Separation Technology, Department of Laboratory Medicine, Semmelweis University, using a Shimadzu Nexera X2 CL-LCMS-8060 CL clinical ultra-performance liquid chromatography-tandem mass spectrometry system and the LabSolutions CL (v1.20) clinical instrument control and data acquisition software (Simkon Kft., Budapest, Hungary). API and RIV were determined simultaneously. The chromatographic separation employed a Phenomenex Kinetex XB-C18 (50 × 2.1 mm, 1.7 µm) column as the stationary phase (kept at 40 °C). The mobile phase consisted of water (A) and methanol (B), both containing 0.1% formic acid. The following gradient program was used (% mobile phase B): 0.00 min (30%); 0.50 min (30%); 3.00 min (50%); 3.01 min (90%); 5.00 min (90%); 5.01 min (30%). The run time was 7.00 min, the mobile phase flow rate was 0.25 mL/min, the autosampler tray was kept at 15 °C, and the injected sample volume was 1.0 µL. A mixture of methanol and water (1:1 *v*/*v*%) was used for rinsing the needle port. The mass spectrometer was operated in the multiple reaction monitoring (MRM) mode following positive electrospray ionization (ESI). The following general settings were employed: nebulizing gas, 3 L/min, drying gas, 6 L/min, heating gas, 12 L/min, desolvation line temperature, 250 °C, heat block temperature, 400 °C, interface temperature, 300 °C. The evaluated ion transitions of API, ^13^C,^2^H_8_-API, RIV, and ^13^C_6_-RIV were 459.90 → 443.10, 468.80 → 199.10, 435.70 → 145.00, and 441.70 → 145.00, respectively. Analyte/IS peak area ratios were employed for evaluation (the respective ion chromatograms are displayed in Appendix A).

Samples were prepared for analysis by using Phenomenex Impact protein precipitation plates (Gen-Lab Kft., Budapest, Hungary). Fifty µL of the plasma sample, calibrator or control, was added to 200 µL internal standard solution. The plate was shaken on a plate shaker at 1100 rpm for 10 min. Using a positive pressure manifold, the filtrate was collected into a 96-well collection plate containing 2-mL wells with a V-shaped bottom, and 150 µL of the supernatant was diluted with 90 µL of water. VAMS samples were prepared by applying the microsampling devices to the surface of whole blood for 5 s and by allowing them to dry for at least 3 h at ambient temperature and without the use of a desiccant.

LC-MS/MS method development, instrument calibration, technical validation, and the assessment of the relationship between analytical bias and the sample hematocrit were performed using de-identified blood samples (left over from routine laboratory diagnostic tests) collected in phlebotomy tubes impregnated with tripotassium ethylene diamine tetraacetate. Hematocrit was measured using a Sysmex XN-1000™ hematology analyzer (Sysmex Hungary Kft., Budapest, Hungary). The specimens had been stored at 2–8 °C for three days before being allocated for disposal. After being selected for technical validation experiments, de-identification was performed. Specimens employed for plasma LC-MS/MS analysis were centrifuged at 2000 rpm, 10 °C for 7 min. The supernatant was transferred to a polypropylene centrifuge tube and kept at −70 °C before being used. The specimens employed for developing and validating the method for processing VAMS samples were used on the day they were selected.

LC-MS/MS quantitation of API and RIV in plasma was performed by running six-point pooled plasma calibrators spiked with each analyte at 5–1000 ng/mL. These plasma calibrators were stored at −70 °C for a maximum of 2 months. For the analysis of VAMS samples, various types of calibrators containing API and RIV in concentration ranges of 10–1000 ng/mL and 5–500 ng/mL, respectively, were employed. Dried plasma and whole blood calibrators were stored at ambient temperature until the end of the VAMS sample processing cycle. Linear calibration models with 1/concentration^2^ weights were applied.

The recoveries of API and RIV from whole blood spiked at concentrations of 500 ng/mL and 250 ng/mL, respectively, and subsequently dried onto VAMS devices, were determined using the following solvent systems containing ^13^C,^2^H_8_-API (35.0 ng/mL) and ^13^C_6_-RIV (16.7 ng/mL):-methanol,-methanol-water 1:1 (vol/vol%),-methanol-acetonitrile 3:1 (vol/vol%), 1:1 (vol/vol%), or 1:3 (vol/vol%).

VAMS devices were filled with blood, were allowed to dry at ambient temperature, and were sonicated (for 30 min at 40 °C) in 200 µL solvent in a 96-well collection plate with a round bottom. In a separate experiment, sonication time (5 min, 15 min, and 30 min) and bath temperature (ambient and 40 °C) were evaluated by employing methanol-acetonitrile 1:1 (vol/vol%). The impact of a second agitation step was investigated by sonication for 15 min, followed by orbital shaking at ambient temperature (1100 rpm for 50 min). Finally, 50 µL non-aqueous extract was diluted with 120 µL of water. Methanol-water 1:1 (vol/vol%) extracts were centrifuged at 10,000× *g* for 5 min, and 50 µL was diluted with 120 µL a methanol-water 1:4 (vol/vol%) mixture. Experiments were performed in triplicate. The reference solvents contained API and RIV at 500 ng/mL and 250 ng/mL, respectively, as well as the internal standards in the concentrations specified above, and were diluted for analysis as described for the extracts.

In the extraction optimization experiments, relative recoveries were calculated using Equation (1):(1)$$Relative\;recovery\;\left(\%\right)=100\times \frac{{analyte\;peak\;area}_{extract}\times {IS\;peak\;area}_{solvent}}{{IS\;peak\;area}_{extract}\times {analyte\;peak\;area}_{solvent}}$$
where *analyte peak area_extract_* refers to the peak area of apixaban or rivaroxaban, while *IS peak area_extract_* corresponds to the peak area of the internal standard in the dried whole blood extracts, *analyte peak area_solvent_* is the peak area of apixaban or rivaroxaban, and *IS peak area_solvent_* is the peak area of the internal standard in the neat solvent system used as a reference.

The technical validation of the LC-MS/MS method was based on the guidelines of the European Medicine Agency, effective from February 2012 [with the only exception that between-run accuracy and precision (A and P) experiments were performed by running three quality control (QC) levels] [14]. Selectivity was assessed by assaying six independent plasma samples that did not contain the analytes or the internal standards (the first step of sample preparation was affected by adding 200 µL acetonitrile to a 50 µL sample). The specificity of the assay was confirmed by running two batches of the same plasma samples. In the first batch, deproteinization was performed by adding 200 µL IS solution to 50 µL plasma. In the second batch, 200 µL acetonitrile was added to 50 µL of the highest-level calibrator. Sample carry-over was assessed by running three cycles of alternate injections of the highest-level calibrator and a blank sample. In all three types of experiments, the areas of the peaks appearing at the retention times of the analytes and the internal standards in their ion chromatograms were required not to exceed 20% and 5%, respectively, of those obtained in the lowest-level calibrator injected in the same run. The lower limits of quantitation corresponded to analyte concentrations at the low end of the calibrated ranges.

Within-run accuracy and precision (A and P) were evaluated in QC samples prepared by spiking API and RIV to independent blank plasma samples (PQC, n = 8), as well as to VAMS samples (WQC, n = 6), at six concentration levels (PQC1–PQC6 and WQC1–WQC6, respectively). Between-run A and P were assessed by re-assaying PQC2, PCQ3, PQC5, WQC2, WQC3, and WQC5 on two separate days. The re-evaluation of three QC levels had been deemed satisfactory considering the fact that in clinical laboratory practice, patient sample batches are very rarely re-run on different days. To confirm the acceptable repeatability of the assay, eight independent plasma matrices were spiked at six concentration levels, generating twice as many QC samples as those recommended by the guideline (six independent samples spiked at four concentration levels) [14]. A and P experiments with VAMS samples were performed by employing dried whole blood, dried plasma, and liquid plasma calibrators. Accuracy was calculated as the mean ratio of measured and nominal concentrations, while precision corresponded to the relative standard deviation (RSD) of these ratios.

IS-corrected matrix factors (IMF) were assessed in plasma by preparing independent blank samples for analysis (n = 6), and then by spiking 5 µL acetonitrile solution of API, RIV, ^13^C, ^2^H_8_-API, and ^13^C_6_-RIV to 140 µL of supernatant. The final mixtures contained the analytes at 1.25 ng/mL (low level) or 50 ng/mL (high level), as well as the internal standards at 31.2 ng/mL (API) and 17.3 ng/mL (RIV). To establish IMF in VAMS samples, independent blank whole blood specimens (n = 6) were applied to VAMS^®^ devices, subsequently dried for 3 h, and sonicated in 200 µL acetonitrile-methanol 1:1 (vol/vol) for 30 min. This was followed by spiking a 5-µL mixture containing API (5 ng/mL or 50 ng/mL), RIV (2.5 ng/mL or 25 ng/mL), ^13^C,^2^H_8_-API (280 ng/mL), and ^13^C_6_-RIV (90.8 ng/mL) to the extracts, 50 µL of which were subsequently diluted with 120 µL water. The stability of the analytes in plasma was tested at three concentration levels by spiking plasma stored at −70 °C for 3 months (n = 5 at each level). Analyte stability was evaluated in VAMS samples by comparing the results obtained in the second between-run A and P assay (performed after storing the samples at ambient temperature for seven days) with the nominal concentration levels (n = 6 at each level). IMF was calculated by dividing the analyte/IS peak area ratio obtained in the prepared blood sample with the ratio observed in the neat solution.

To evaluate the impact of sample hematocrit, two experiments were performed, with ten independent blood samples (n = 5 in each experiment) being processed separately after hematocrit measurement. In the first experiment, two aliquots of each blood sample with volumes of 600 µL and 800 µL (hematocrits: 0.40–0.46 L/L) were pipetted into 1.5-mL polypropylene tubes. The 800-µL aliquot was centrifuged at 1000× *g* before removing 260 µL plasma (final hematocrit: 0.59–0.68 L/L) subsequently added to the 600-µL aliquot (final hematocrits: 0.28–0.32 L/L). In the second experiment, one 500-µL and two 650-µL aliquots were prepared. Plasma (150 µL, 215 µL) was removed from the two 650-µL aliquots (final hematocrits: 0.52–0.55 L/L and 0.61–0.63 L/L, respectively). Subsequently, 90 µL plasma was added to the 500-µL aliquot (final hematocrit: 0.34–0.36 L/L). Each preparation was homogenized by gentle manual rotation. Two 190-µL aliquots were transferred to 1.5-mL polypropylene tubes. The analytes were spiked to these by adding 10 µL of a methanolic solution containing 2000 ng/mL API and 1000 ng/mL RIV (low-level), or 10,000 ng/mL API and 5000 ng/mL RIV (high-level). The samples were homogenized by gentle manual rotation before their application to the VAMS^®^ devices. Six-point dried plasma calibrators containing API and RIV at 10–1000 ng/mL and 5500 ng/mL, respectively, were employed.

The relationship between the relative recovery (*RR*) of each analyte and the sample hematocrit was modeled mathematically in dried whole blood as described in Appendix B. In the cross-validation experiments, API and RIV concentrations measured in VAMS samples were corrected using Equations (2) and (3) (see Appendix B for the details of constructing the equations).
(2)$$\begin{array}{c} {RR}_{API}=Hct\times \left(-0.000787\cdot c_{API}-0.6417\right)\\ +{0.9608+0.006786\times \tan}^{-1}\left|-0.000787\cdot c_{API}-0.6417\right| \end{array}$$
and
(3)$$\begin{array}{c} {RR}_{RIV}=Hct\times \left(-0.002173\cdot c_{RIV}-0.3981\right)\\ +{0.9145+0.008716\times \tan}^{-1}\left|-0.002173\cdot c_{RIV}-0.3981\right| \end{array}$$
where *RR_API_* and *RR_RIV_* are the relative recoveries of API and RIV, Hct is the hematocrit (L/L), while *c_API_* and *c_RIV_* are the measured concentrations of API and RIV, respectively.

Method cross-validation was accomplished by employing 60 whole blood samples, each collected in 3.5-mL blue-top phlebotomy tubes containing 0.3 mL 3.2% trisodium citrate, and admitted by the Central Laboratory, Department of Laboratory Medicine, Semmelweis University (Budapest, Hungary) for laboratory diagnostic tests including whole blood count and the monitoring of API (n = 21) or RIV (n = 39) concentrations (Greiner Bio-One no. 484518). No patient data were collected, and no interaction was made with the donors of the samples. Due to the strictly technical nature of this study, no ethical approval was sought.

After performing a whole blood count using the primary tube, a 0.5-mL aliquot of each sample was transferred to a secondary 1.5-mL polypropylene tube, was de-identified, and was assigned a numerical identifier displayed on both tubes as well as on the VAMS^®^ devices. Hematocrits were recorded manually on the secondary tubes. VAMS samples were prepared by contacting the surface of the blood in the secondary tube for 5 s, and drying at ambient temperature. The dried samples were stored on the bench without using a desiccant until the day of analysis.

The primary and secondary tubes were centrifuged subsequently at 2000× *g*, 10 °C for 7 min. The anti-Xa chromogenic assay of API or RIV was performed in the plasma separated in the primary tube on a Siemens BCS^®^ XP (Siemens Healthcare Diagnostics Inc., Tarrytown, NY, USA) automated hemostasis analyzer platform using the Siemens Innovance^®^ BCS^®^ Anti-Xa (Siemens Healthcare Diagnostics Inc., Tarrytown, NY, USA) one-step chromogenic assay. Calibration was accomplished using Hyphen Bio-Med (Neuville-sur-Oise, France) API and RIV calibrators (reference numbers: 226201 and 222701, lot numbers: FA138119 and FA229519, respectively). Calibrators were stored as recommended by the manufacturer. Three-point linear calibration models were constructed without weights in concentration ranges of 0–663 ng/mL and 0–500 ng/mL for API and RIV, respectively. For controlling the assay, Hyphen Bio-Med API and RIV bi-level internal controls (reference numbers: 225201 and 224501, lot numbers: FB15371F and FB01061B, respectively) were employed (Siemens Healthcare Kft., Budapest, Hungary). Two hundred microliters of plasma separated in the secondary tube were transferred to another polypropylene container and kept at −70 °C until LC-MS/MS analysis.

A sample processing cycle spanning seven days was evaluated for managing VAMS samples prepared for the cross-validation experiment. Cycles began with the arrival of the first sample (day 0) and ended with the analysis of samples admitted over seven days. Quantitation was performed using whole blood spiked with API and RIV dried onto VAMS^®^ devices on days 0, 3, or 7; plasma spiked and dried onto VAMS^®^ devices on day 3; as well as liquid plasma stored at −70 °C and thawed on the day of analysis as calibrators (Figure 1).

Violin plots showing the agreement between API and RIV concentrations measured in VAMS and in plasma samples using LC-MS/MS were generated after trimming the lowest and highest RR values (corresponding to the ratio of analyte concentrations in plasma and dried whole blood) using the *ggplot2* package of the computing platform R (version 4.2.2) [15]. The results of cross-validation experiments were evaluated by Bland-Altman analysis using a Microsoft Excel spreadsheet prepared by the authors. In addition, Passing-Bablok regression was performed using the *mcr* package of R. Linearity of the relationship between the datasets compared was established by calculating the h-statistic as recommended by Passing and Bablok (the confidence level was 95%, and the corresponding critical value was 1.36) [16]. The visualization of the regression plots was done by implementing an R script published on the website https://rowannicholls.github.io/R/statistics/agreement/passing_bablok.html (accessed on 6 July 2024, see Appendix A) with some modifications. Lin’s concordance correlation coefficients (CCC) were calculated using the *DescTools* package.

## 3. Results

### 3.1. Relative Recovery of the Analytes from VAMS Samples

The mean RR of API from VAMS samples was 89.6–100% when methanol, or mixtures of methanol and acetonitrile with up to 50% acetonitrile content, were used as the extraction solvent. Mean RR values of 70.0% were obtained for API when methanol-water 1:1 or methanol acetonitrile 1:3 was employed. The mean RR of RIV was 101.6–114.6% when acetonitrile-methanol mixtures were applied. The application of methanol-water 1:1 (vol/vol%) as the extraction solvent yielded a mean RR of 76.3%. Eventually, the solvent of choice was methanol-acetonitrile 1:1 (vol/vol%). Sonication with or without subsequent orbital shaking proved equally efficient. The duration of sonication or shaking had a considerable effect on the relative recoveries. Elevated bath temperature had no impact on recoveries. In favor of its simplicity, sonication for 30 min at ambient temperature was selected for further application (Figure 2).

### 3.2. Technical Validation of the LC-MS/MS Assays

The LC-MS/MS method developed allowed the selective and specific detection of API, RIV, and the internal standards in plasma and the extracts of VAMS samples, with no interfering peaks or sample carry-over observed. The mean accuracy of the plasma assay (with RSD ranges in parentheses) was 94.4–114.0% (3.9–12.8%) and 98.1–111.7% (1.7–6.9%) for API and RIV, respectively. When processing VAMS samples, the mean accuracy of the API assay was 90.6–99.3% (2.6–8.0%), 94.6–102.6% (2.6–7.8%), and 105.4–121.7% (2.6–8.7%) when dried whole blood, dried plasma, or liquid plasma calibrators were used, respectively. For RIV, the mean accuracy was 92.3–100.7% (1.4–10.9%), 92.3–100.2% (1.4–11.0%), and 116.5–126.2% (1.4–12.1%), respectively (Table 1). The reason for obtaining irrationally high recoveries when liquid plasma calibrators were used is assumed to be a difference in matrix effects (while liquid plasma calibrators were diluted eightfold by the end of sample preparation, VAMS samples were diluted 34-fold). In plasma, the mean IS-corrected matrix factors of API and RIV were 0.86 ± 0.05 and 0.90 ± 0.13 at low concentration levels, respectively. In all other experimental setups, the IMF showed only negligible deviations from 1.00 (Table 2). The relative recoveries of API and RIV after storing plasma samples at −70 °C for 3 months were 89.5–118% and 96.1–109%, respectively. When VAMS samples were kept at ambient temperature, relative recoveries of 91.3–110% and 90.1–106% were obtained after 7 days, respectively (Table 3).

### 3.3. Hematocrit- and Concentration-Dependence of the Relative Analyte Recovery from VAMS Samples

Mean slopes of −0.710 and −0.500 were obtained at low concentrations, as well as of −0.959 and −0.877 at high concentrations for API and RIV, respectively, when a linear relationship between the sample hematocrit and the relative recoveries was modeled. The median determination coefficients of the individual slopes were 0.939 (range: 0.051–0.996) and 0.906 (0.036–1.000) at the low, as well as 0.981 (0.472–0.999) and 0.954 (0.545–0.998) at the high spiking levels, respectively (details are provided in Appendix A). The differences between the slopes obtained at the low and high spiking levels indicate that the RRs of these analytes are a function of both the hematocrit and the concentration. The direction of the trend in RR was the same in 80% of the individual samples (Figure 3A,B). The correction of the analyte concentrations measured by employing Equations (2) and (3) improved the agreement with the nominal concentrations considerably (Figure 3C–F).

### 3.4. Cross-Validation Experiments Using Leftover Patient Samples

The median VAMS/plasma API concentration ratios obtained using the integrated LC-MS/MS method showed good agreement, regardless of the type of calibrator employed, with the dispersion of concentration ratios being smaller when dried plasma was used. For RIV, the median concentration ratios displayed best agreement when dried plasma (prepared on day 3 of the 7-day sample processing cycle) or dried whole blood calibrators (prepared on day 0 or day 3) were employed, with the smallest dispersion of differences being observed for results obtained by performing quantitation using dried plasma calibrators (Figure 4). A systematic bias of RIV concentration ratios was observed when the quantitation in VAMS samples was conducted using liquid plasma or dried whole blood calibrators prepared on day 7.

Bland-Altman analyses demonstrated that consistently lower API and RIV plasma concentrations were obtained using the LC-MS/MS method in comparison to the anti-Xa chromogenic assay [systematic bias: −27.2% (95% confidence interval: −87.7–33.3%) and −33.8% (95% confidence interval: −71.5–3.8%, respectively]. In contrast, the systematic bias of plasma concentrations obtained using LC-MS/MS versus those measured in VAMS samples was acceptable [LC-MS/MS vs. VAMS with dried plasma calibrators: −0.9% (95% confidence interval: −84.3–82.6%) and −2.4% (95% confidence interval: −59.0–54.3%, respectively; LC-MS/MS vs. VAMS with whole blood calibrators dried onto VAMS devices on day 0: −0.5% (95% confidence interval: −81.3–82.3%) and –3.6% (95% confidence interval: −53.3–46.1%, respectively]. In VAMS samples, the agreement between API and RIV concentrations obtained by using dried plasma calibrators or by using dried whole blood calibrators prepared on day 0 of the 7-day sample processing cycle was verified. The systematic bias was 1.3% (95% confidence interval: −17.9%–0.5%) and −1.3% (95% confidence interval: −29.7%–27.1%) for API and RIV, respectively (Figure 5). There was a single outlier in the comparative measurements of RIV; after its exclusion, the sytematic bias improved to 0.7% (95% confidence interval: −14.1–15.4%).

The results of the Passing-Bablok regression are shown in Figure 6 and Table 4. The linearity of the relationship between the data compared was confirmed in all comparisons, with the h statistic being lower than the critical value (1.36). For both analytes, the slope of the regression line was statistically different from 1 when comparing the results obtained in plasma using LC-MS/MS and the anti-Xa assay, indicating a lack of equivalence due to proportional error (bias). The intercepts were not different from 0 statistically, confirming that no source of constant error could be associated with the disagreement observed. The Kendall’s τ correlation coefficients of the regression lines were 0.826–0.943 and 0.880–0.949 for API and RIV, respectively, and exceeded 0.90 only when comparing the results obtained in VAMS samples obtained by calibration using dried plasma and dried whole blood (the latter prepared on day 0 of the sample processing cycle). Based on McBride’s evaluation, Lin’s concordance correlation coefficient indicated a weak agreement between the plasma LC-MS/MS and anti-Xa RIV assays (CCC < 0.90), moderate agreement between the plasma LC-MS/MS and anti-Xa API assays, as well as between RIV concentrations obtained in plasma and in VAMS samples (0.90 < CCC < 0.95), substantial agreement between API concentrations obtained in plasma and in VAMS samples (0.95 < CCC < 0.99), and almost perfect agreement between results obtained in VAMS samples using dried plasma or dried whole blood calibrators (the latter prepared on day 0 of the sample processing cycle, CCC > 0.99) [17]. Data underlying the comparisons are provided as Appendix A.

## 4. Discussion

To the knowledge of the authors, the first integrated LC-MS/MS methodology is presented for the quantitation of API and RIV in plasma and whole blood VAMS samples. There are earlier examples of using dried blood spots (DBS) collected onto Whatman 903 cards, the limitations of which include lower and more variable recoveries [18,19,20,21,22,23]. VAMS is a more advanced technique, characterized in general by excellent and highly reproducible analyte recoveries [24].

Employing VAMS devices impacts laboratory assay costs, especially if the number of patient samples in the batch is low. The 7-day sample processing cycle implemented to increase cost-efficiency has yielded accurate and reproducible assay results. Calibration with whole blood dried on day 0 or day 3, as well as with plasma dried on day 3 of the cycle, gave rise to accurate results, in contrast to whole blood dried on the day of analysis or to liquid plasma. In the A and P experiments, the relative standard deviations of the mean accuracies obtained for the three types of calibrators were almost identical. Employing liquid plasma calibrators led to the overestimation of concentrations (likely due to the substantial differences in sample dilution during sample preparation: liquid plasma was diluted eightfold, while dried blood samples were diluted 34-fold), whereas mean accuracies obtained using dried whole blood or dried plasma calibrators were close to 100% and in good agreement. In the cross-validation experiments, the results obtained using dried plasma and dried whole blood calibrators (the latter prepared on day 0 of the sample processing cycle) were in excellent agreement, as demonstrated by Bland-Altman analysis, Passing-Bablok regression, and the calculation of Lin’s concordance correlation coefficients. These results indicate that, when available, commercially lyophilized calibrator and control products employed for plasma assays may be applicable to LC-MS/MS dried capillary blood sample analysis as well, which increases the reliability of the assays.

The complex relationship between the hematocrit, the analyte concentration, and the relative recovery has been modeled successfully for both API and RIV. Similar approaches have been proposed for protein kinase inhibitors [25,26]. We conclude that establishing the hematocrit of each dried whole blood microsample is required, e.g., by measuring potassium concentrations of the hemolysate in VAMS extracts [27,28].

The results obtained in plasma and VAMS samples using the integrated LC-MS/MS method displayed discordance due to random differences, though without any sign of bias. Since care was taken to homogenize blood before preparing the VAMS samples, and to soak blood onto the VAMS device in a standard procedure, the discrepancies are attributed to non-technical causes, such as partitioning between plasma and red blood cells. In plasma, a positive systematic bias of the automated anti-Xa chromogenic assay was observed, along with a considerable dispersion of relative differences from LC-MS/MS results. Passing-Bablok regression confirmed that the proportional bias was especially remarkable in the case of RIV, while no constant bias was identified for either analyte. In a large prospective, multi-center cross-sectional study, strong correlation but no analytical equivalence was found between the Siemens anti-Xa assay and another LC-MS/MS method [29]. In another work, strong correlations between API and RIV concentrations were obtained in matched samples, but along with confidence intervals up to ±60% and ±76.8% of the mean relative differences, respectively [30]. Excellent agreement between the results obtained using LC-MS/MS and other commercial anti-Xa assays has been reported [31,32]. At the same time, systematic discrepancies were observed between an LC-MS/MS method and the STAGO STA-Liquid Anti-Xa chromogenic assay [33,34]. Such differences in the concordance of clinical laboratory assays are not unusual and underpin the importance of inter-laboratory quality assessment schemes.

## 5. Conclusions

Our findings demonstrate that the outcomes of API and RIV assays depend considerably on the approach selected for sampling and analysis. The results obtained in plasma and VAMS samples using LC-MS/MS were not equivalent, despite employing an integrated LC-MS/MS methodology, showing that sample type itself is a crucial factor. In addition, the commercial automated anti-Xa chromogenic assay failed to yield API and RIV plasma concentrations equivalent to those obtained by using LC-MS/MS. Our findings show that the accurate ongoing documentation of the hemostatic function and the pharmacotherapy applied (including the administration of heparin) is required for the correct interpretation of API and RIV TDM results, as any phenomenon that affects the coagulation cascade may also distort the results of anti-Xa measurements. It is imperative that the sampling procedure be standardized, and the use of the type of blood collection tube defined by the assay kit manufacturer, as well as the collection of the correct volumes of blood, is pivotal for minimizing systematic and random measurement errors.

Importantly, all three approaches discussed herein have unique clinical advantages. The analytical discrepancies must be taken into account. Clinical professionals should be aware of the type of sample processed and the technology employed for each API and RIV measurement to interpret results correctly. This information should therefore be reported by clinical laboratories along with TDM results.

## Figures and Tables

**Figure 1 diagnostics-14-01939-f001:**
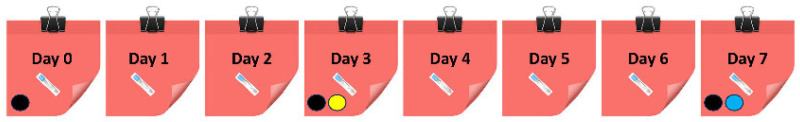
The seven-day sample processing cycle designed for the cost-efficient management of dried whole blood volumetric absorptive microsamples prepared for the analysis of apixaban and rivaroxaban. Patient samples were admitted for the therapeutic monitoring of either drug on days 0 thru 7, indicated by (
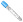
). (●) shows the preparation of dried whole blood calibrators, (●) marks the preparation of dried plasma calibrators, and (●) displays the application of liquid plasma calibrators.

**Figure 2 diagnostics-14-01939-f002:**
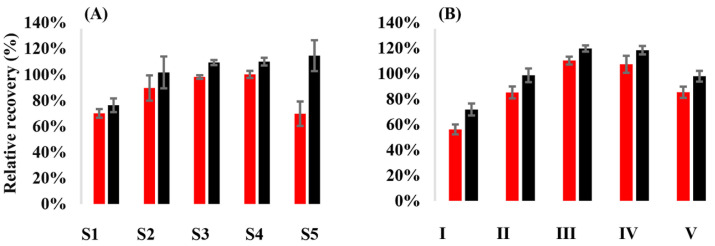
Internal standard-corrected relative recoveries of apixaban and rivaroxaban from dried whole blood volumetric absorptive blood microsamples (n = 3) using (**A**) various extraction solvents and a uniform extraction process, and (**B**) using acetonitrile-methanol 1:1 (vol/vol%) as the extraction solvent and various extraction processes. Red bars show results obtained for apixaban, while black bars display relative recoveries of rivaroxaban. The extraction solvents were S1—methanol-water 1:1 (vol/vol%), S2—methanol, S3—acetonitrile-methanol 1:3 (vol/vol%), S4—acetonitrile-methanol 1:1 (vol/vol%), and S5—acetonitrile-methanol 3:1 (vol/vol%). The extraction settings were: I—sonication at ambient temperature for 5 min, II—sonication at ambient temperature for 15 min, III—sonication at ambient temperature for 30 min, IV—sonication in a water bath pre-heated to 40 °C for 15 min, and V—sonication at ambient temperature for 15 min, followed by orbital shaking at 1100 rpm for 50 min. Experiments (**A**,**B**) were performed on separate days.

**Figure 3 diagnostics-14-01939-f003:**
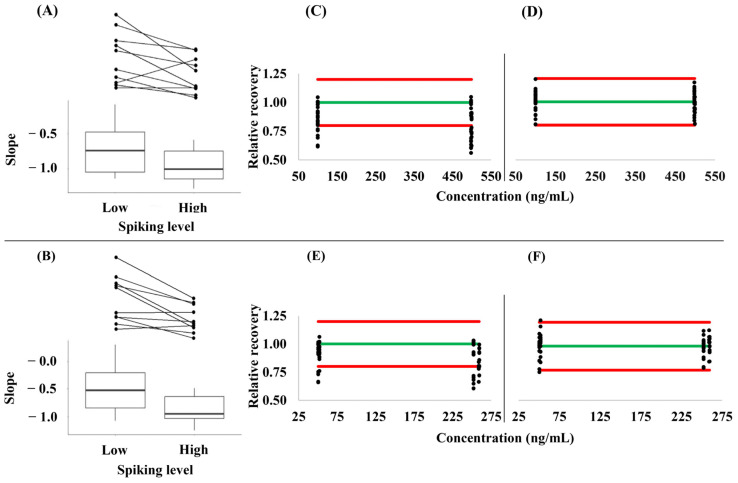
(**A**,**B**) Slopes of the linear relationship between the sample hematocrit and the relative recovery of the analytes from dried volumetric absorptive microsamples at low and high spike levels of (**A**) apixaban and (**B**) rivaroxaban. (**C**–**F**) The impact of the correction of the concentrations measured in 10 independent blood samples at the two spike levels each by using Equations (1) and (2). (**C**) Uncorrected relative recovery of apixaban. (**D**) corrected relative recovery of apixaban. (**E**) uncorrected relative recovery of rivaroxaban. (**F**) corrected relative recovery of rivaroxaban. The green line (**–**) corresponds to perfect agreement (relative recovery = 1.0), while the red lines (**–**) indicate the boundaries of the accepted range (0.8–1.2) of the relative recovery.

**Figure 4 diagnostics-14-01939-f004:**
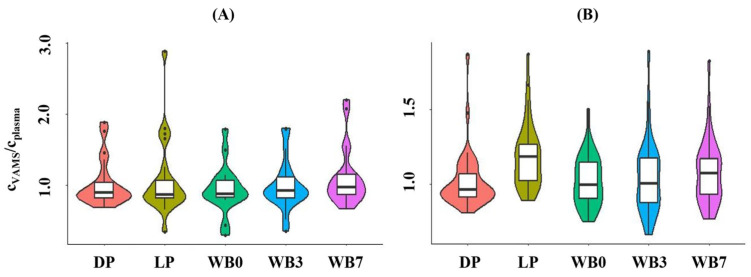
Violin plots showing the concentration ratios of (**A**) apixaban and (**B**) rivaroxaban obtained in dried whole blood volumetric absorptive whole blood microsamples (VAMS) and in the plasma of patients by using various types of calibrators for quantitation in VAMS. Dried calibrators were prepared by applying whole blood or plasma spiked with the analytes to VAMS^®^ devices. C_plasma_ concentration measured in plasma using LC-MS/MS. c_VAMS_, concentration measured in the VAMS sample, and corrected by applying Equation (1) or Equation (2). DP, dried plasma prepared on day 3 of the weekly sample processing cycle. LP, liquid plasma. WB0, whole blood spiked and dried on day 0 of the sample processing cycle. WB3, whole blood spiked and dried on day 3 of the sample processing cycle. WB7, whole blood spiked and dried on day 7 of the sample processing cycle.

**Figure 5 diagnostics-14-01939-f005:**
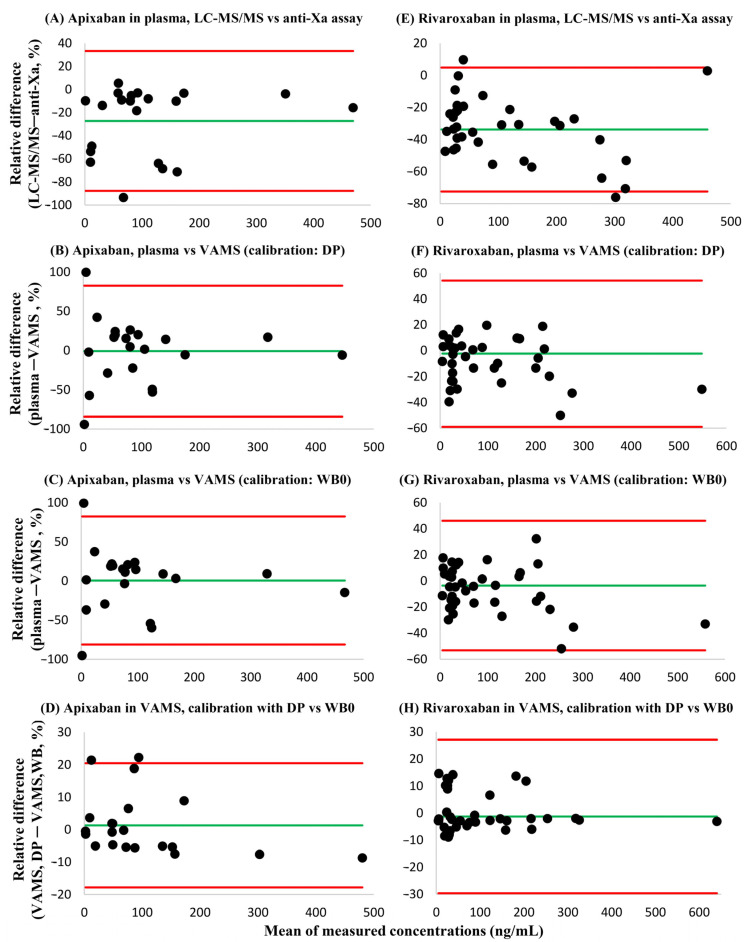
Bland-Altman plots showing the bias and dispersion of the relative differences of apixaban and rivaroxaban concentrations found in blood samples using the analytical approaches described. Green lines (**–**) represent mean relative differences, while red lines (**–**) display the 95% confidence intervals. (**A**) Apixaban, plasma LC-MS/MS versus plasma anti-Xa assay results. (**B**) Apixaban, results obtained in plasma employing LC-MS/MS and in dried whole blood volumetric absorptive microsamples (VAMS), the latter determined by using dried plasma calibrators (DP) prepared on day 3 of the seven-day sample processing cycle. (**C**) Apixaban, results obtained in plasma using LC-MS/MS, and in VAMS samples using dried whole blood (WB0) calibrators prepared on day 0 of the cycle. (**D**) Apixaban, results obtained in VAMS samples by using DP and WB0 calibrators. (**E**) Rivaroxaban, plasma LC-MS/MS versus plasma anti-Xa assay results. (**F**) Rivaroxaban, results obtained in plasma (LC-MS/MS), and in VAMS samples using DP calibrators. (**G**) Rivaroxaban, results obtained in plasma (LC-MS/MS), and in VAMS samples using WB0 calibrators. (**H**) Rivaroxaban, results obtained in VAMS samples by using DP versus WB0 calibrators.

**Figure 6 diagnostics-14-01939-f006:**
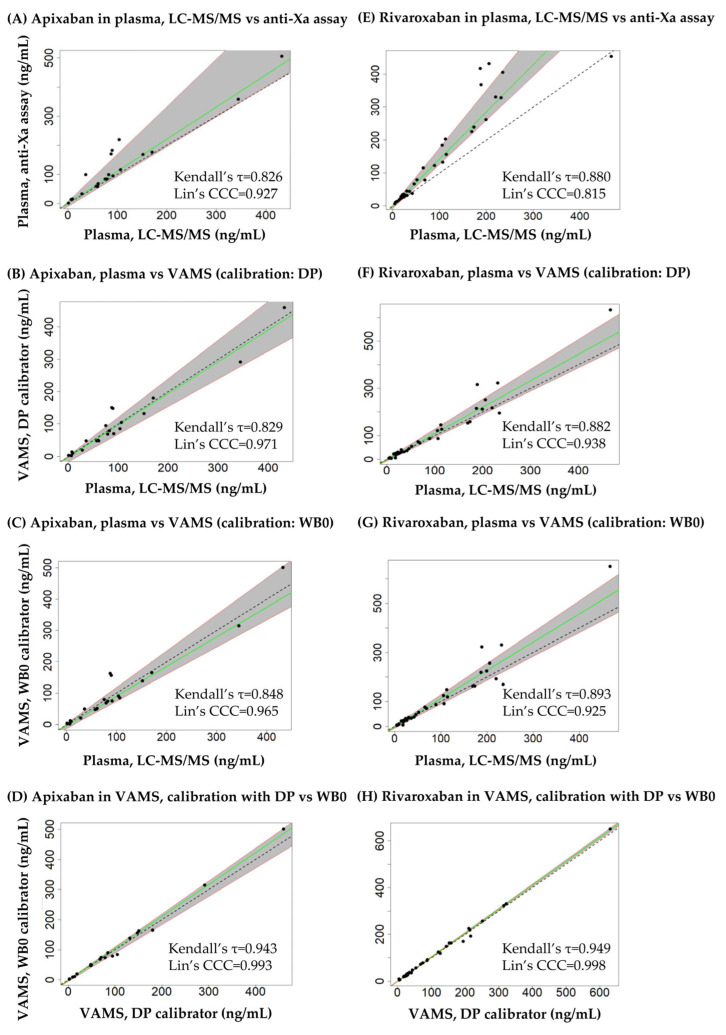
Passing-Bablok regression plots of apixaban and rivaroxaban concentrations observed in blood samples using the analytical approaches described in the manuscript. The plots also show Lin’s concordance correlation coefficients (Lin’s CCC). (**A**) Apixaban plasma liquid chromatography-tandem mass spectrometry (LC-MS/MS) versus plasma anti-Xa assay results. (**B**) Apixaban, results obtained in plasma employing LC-MS/MS and in dried whole blood volumetric absorptive microsamples (VAMS), the latter determined by using dried plasma calibrators (DP) prepared on day 3 of the seven-day sample processing cycle. (**C**) Apixaban, results obtained in plasma using LC-MS/MS and in VAMS samples using dried whole blood (WB0) calibrators prepared on day 0 of the cycle. (**D**) Apixaban, results obtained in VAMS samples by using DP and WB0 calibrators. (**E**) Rivaroxaban, plasma LC-MS/MS versus plasma anti-Xa assay results. (**F**) Rivaroxaban, results obtained in plasma (LC-MS/MS), and in VAMS samples using DP calibrators. (**G**) Rivaroxaban, results obtained in plasma (LC-MS/MS), and in VAMS samples using WB0 calibrators. (**H**) Rivaroxaban, results obtained in VAMS samples by using DP versus WB0 calibrators.

**Table 1 diagnostics-14-01939-t001:** Results of the accuracy and precision study of the liquid chromatography-tandem mass spectrometry assays conducted in plasma and in dried whole blood volumetric absorptive microsamples (VAMS). RSD, relative standard deviation. QC, quality control sample.

Performance Characteristics	Within-Run Experiment						Between-Run Experiment, Day 1			Between-Run Experiment, Day 2		
	QC 1	QC 2	QC 3	QC 4	QC 5	QC 6	QC 2	QC 3	QC 5	QC 2	QC 3	QC 5
Apixaban in plasma												
Accuracy (%)	99.9	105.3	95.6	103.5	114.0	111.3	102.1	98.1	104.4	97.2	94.4	104.1
RSD (%)	4.9	4.3	12.8	6.8	4.1	3.9	2.8	5.2	5.4	5.1	6.2	5.6
Apixaban in VAMS, whole blood calibrators												
Accuracy (%)	94.1	94.7	93.8	91.7	95.8	90.6	99.3	95.9	96.9	97.0	94.0	98.3
RSD (%)	8.0	6.5	4.8	7.8	3.4	7.1	4.8	5.3	2.6	5.1	6.3	4.7
Apixaban in VAMS, dried plasma calibrators												
Accuracy (%)	100.0	98.2	96.9	94.6	98.8	93.4	102.8	99.4	100.5	102.3	98.6	102.6
RSD (%)	7.7	6.5	4.8	7.8	3.4	7.1	4.8	5.3	2.6	5.0	6.3	4.7
Apixaban in VAMS, liquid plasma calibrators												
Accuracy (%)	105.4	114.5	114.3	112.4	117.7	111.4	115.8	113.0	115.2	120.6	116.6	121.7
RSD (%)	8.7	6.6	4.8	7.8	3.4	7.1	4.9	5.4	2.6	5.1	6.3	4.7
Rivaroxaban in plasma												
Accuracy (%)	98.1	103.8	107.8	105.7	111.7	107.0	101.2	101.2	104.4	100.5	100.1	100.8
RSD (%)	3.8	4.3	2.2	1.9	2.4	1.7	6.9	3.3	5.7	4.5	3.3	1.7
Rivaroxaban in VAMS, whole blood calibrators												
Accuracy (%)	100.7	96.6	97.7	93.3	97.1	92.3	97.8	96.1	97.2	94.8	93.6	98.5
RSD (%)	10.9	4.7	4.6	5.6	2.9	6.1	4.7	3.6	1.4	8.1	5.9	3.5
Rivaroxaban in VAMS, dried plasma calibrators												
Accuracy (%)	100.2	96.5	97.7	93.3	97.2	92.3	97.8	96.2	97.3	94.5	92.4	96.4
RSD (%)	11.0	4.7	4.6	5.6	2.9	6.1	4.7	3.6	1.4	7.9	5.8	3.5
Rivaroxaban in VAMS, liquid plasma calibrators												
Accuracy (%)	116.5	121.3	124.0	119.1	124.3	118.2	117.5	117.4	120.4	126.2	122.1	126.2
RSD (%)	12.1	4.7	4.6	5.7	2.9	6.1	4.9	3.6	1.4	7.8	5.7	3.5

**Table 2 diagnostics-14-01939-t002:** Internal standard-corrected matrix factors affecting the analysis of apixaban (API) and rivaroxaban (RIV). Means and standard deviations are shown, with the relative standard deviations provided in parentheses.

Matrix	API		RIV	
	Low Spiking Level	High Spiking Level	Low Spiking Level	High Spiking Level
Plasma	0.86 ± 0.05 (5.4%)	1.01 ± 0.07 (7.2%)	0.90 ± 0.13 (15.0%)	1.05 ± 0.06 (6.1%)
Dried whole blood	1.00 ± 0.04 (3.6%)	0.96 ± 0.05 (5.5%)	1.03 ± 0.05 (4.9%)	0.94 ± 0.06 (6.7%)

**Table 3 diagnostics-14-01939-t003:** Results of the plasma and dried whole blood storage tests. Plasma samples (n = 5) were stored at −70 °C for 3 months, dried whole blood samples (n = 6) were stored at ambient temperature for 7 days. API, apixaban. RIV, rivaroxaban.

Analyte	Spiking Level	Spiked Concentration (ng/mL)	Relative Recovery (%)	
			Plasma	VAMS Samples
API	Low	53.2	94.3–118	
		49.8		98.0–110
	Middle	107	93.4–113	
		99.6		91.3–105
	High	428	89.5–103	
		498		96.3–105
RIV	Low	53.2	96.1–106	
		25.2		94.0–106
	Middle	107	102–109	
		50.4		90.1–100
	High	428	100–107	
		252		95.1–99.2

**Table 4 diagnostics-14-01939-t004:** Results of the Passing-Bablok regression analysis. CI, confidence interval. DP, dried plasma calibrators prepared on day 3 of the seven-day sample processing cycle. LC-MS/MS, liquid chromatography-tandem mass spectrometry. VAMS, volumetric absorptive microsample. WB0, dried whole blood calibrators prepared on day 0 of the seven-day sample processing cycle.

Substance	Approaches Compared	Slope (95% CI)	Intercept (95% CI)	h Statistic
Apixaban	Plasma, LC-MS/MS and plasma, Anti-Xa	1.11 (1.03–1.59)	0.12 (−12.7–5.73)	0.91
	Plasma, LC-MS/MS and VAMS (calibrator: DP)	0.98 (0.86–1.23)	−2.26 (−18.3–1.74)	0.91
	Plasma, LC-MS/MS and VAMS (calibrator: WB0)	0.95 (0.87–1.16)	−4.25 (−13.6–1.80)	1.21
	VAMS, DP calibrator and WB0 calibrator	1.06 (0.94–1.09)	−0.91 (−4.92–0.23)	0.91
Rivaroxaban	Plasma, LC-MS/MS and plasma, Anti-Xa	1.43 (1.34–1.75)	−1.14 (−8.39–0.85)	0.89
	Plasma, LC-MS/MS and VAMS (calibrator: DP)	1.12 (0.97–1.26)	−2.27 (−7.20–1.76)	0.89
	Plasma, LC-MS/MS and VAMS (calibrator: WB0)	1.15 (0.97–1.27)	−2.09 (−6.85–1.40)	1.12
	VAMS, DP calibrator and WB0 calibrator	1.02 (1.00–1.03)	−0.05 (−1.21–0.87)	0.89

## Data Availability

Underlying datasets are provided in the Appendix A.

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
