# Peer review of "Assays for Monitoring Apixaban and Rivaroxaban in Emergency Settings, State-of-the-Art Routine Analysis, and Volumetric Absorptive Microsamples Deliver Discordant Results"

_diagnostics, 2024, doi:10.3390/diagnostics14171939_

Round 1
Reviewer 1 Report
Comments and Suggestions for Authors
In the manuscript titled with "Assays applicable for the therapeutic monitoring of apixaban and rivaroxaban concentrations in emergency settings, state-of-the-art routine analysis and dried whole blood volumetric absorptive microsamples collectible by home sampling deliver discordant results", the authors compared two relevant methods on their performance to detect and quantify API and RIV. LC/MS/MS method coupled with VAMS was validated and compared with commercial anti-Xa method. Based on that, the authors were able to introduce the feasibility of this method to test API and RIV in a patient-friendly manner. Overall, the story was well-established, and the science is sound. I would recommend a minor revision before its acceptance according to the comments below:
1. In the introduction part, the authors briefly introduced the necessity of the quantitation of API and RIV, and the traditional LC/MS/MS method vs. VAMS method. However, the introduction of VAMS method from its basic principle was not sufficient. Since VAMS is applied in the method to be developed, additional information needs to be provided to justify the parameters selection during the validation.
2.In the results part 3.1, the authors' selection of diluent was not well rationalized. From the figure 2, S4 should be the best, which is ACN/MeOH = 3:1. Why did the authors finally selected 1:1? Besides, in line 284, the RR value of 1:3 should not be 70%, but 100% based on Figure 2. Also, the ID of red and black bar were not demonstrated.
3.In the validation part, high recovery was observed for liquid plasma calibrators. What is the root cause and what can be the potential prevention? This is critical as it is not a precision issue, but an accuracy issue, which can cause critical problem on result deliver.
4. In section 3.3, the slope difference need to be proved by R square. The authors needs to include it in the main text.
Author Response
- In the introduction part, the authors briefly introduced the necessity of the quantitation of API and RIV, and the traditional LC/MS/MS method vs. VAMS method. However, the introduction of VAMS method from its basic principle was not sufficient. Since VAMS is applied in the method to be developed, additional information needs to be provided to justify the parameters selection during the validation.
The Introduction has been amended to underpin the importance of including the assessment of the impact of hematocrit on analyte recovery in the validation process (lines 60-74).
- In the results part 3.1, the authors' selection of diluent was not well rationalized. From the figure 2, S4 should be the best, which is ACN/MeOH = 3:1. Why did the authors finally selected 1:1? Besides, in line 284, the RR value of 1:3 should not be 70%, but 100% based on Figure 2. Also, the ID of red and black bar were not demonstrated.
We highly appreciate this remark. The reason underlying the discrepancies noted by the Reviewer was that in the caption of Figure 2.A the solvents and solvent compositions which yielded the recoveries shown were listed incorrectly. The details of the extraction yields in the main text, and the decision to use acetonitrile-methanol 1:1 (vol/vol%) in the downstream experiments were correct. Of note, S4 in Figure 2 corresponds to acetonitrile-methanol 1:1 (vol/vol%), which was found to be the optimal composition for our purposes.
The use of red and black colors in Figure 2 is now explained in the caption.
- In the validation part, high recovery was observed for liquid plasma calibrators. What is the root cause and what can be the potential prevention? This is critical as it is not a precision issue, but an accuracy issue, which can cause critical problem on result deliver.
The root cause is assumed to be a difference in matrix effects. While liquid plasma calibrators were diluted eightfold by the end of sample preparation, VAMS samples were diluted 34-fold. This is displayed in lines 335-338.
- In section 3.3, the slope difference need to be proved by R square. The authors needs to include it in the main text.
The determination coefficients requested by the Reviewer have been included in the main text (lines 362-366), and the the data underlying for the calculations are provided in the Supplementary file.

Reviewer 2 Report
Comments and Suggestions for Authors
The authors have presented a comprehensive manuscript with clinicaly relevant work on the analysis of two drugs important for their therapeutic drug monitoring.
Several analytical approchaes have been compared, and some important conclusions for clinical practise have been made.
Some comments are as follows:
1. Title is very lengthy. Consider shortening it. Perhaps the word „concentrations“ can be omitted.
2. What is the meaning of the end of the title?
3. Line 26 – please rephrase the sentence
4. Please consider of putting the validation data into a table. The senteces are very hard to read and follow.
5. Have you obtained Deming regression curves?
6. have you obtained Lin’s concordance correlation coefficients?
7. Please include MRM chromatograms into supplementary data
8. It would be valuble to draw additional conclusion based on the obtained results as it would be valuble for readers, but also for clinical practice.
Comments on the Quality of English Language
English quality is fine. Some minor corrections could be made for the better flow of the sentences.
Author Response
- Title is very lengthy. Consider shortening it. Perhaps the word „concentrations“ can be omitted.
The title has been rephrased as follows: Assays for monitoring apixaban and rivaroxaban in emergency settings, state-of-the-art routine analysis and volumetric absorptive microsamples deliver discordant results.
- What is the meaning of the end of the title?
The end of the title was not displayed by the editorial manager software, but could be seen in full in the manuscript. Shortening the title has hopefully eliminated this issue.
- Line 26 – please rephrase the sentence
The abstract has been updated. As part of this update, the sentence in lines 24-26 has been rephrased, and the duplicated mentioning of the seven-day sample processing cycle has been corrected.
- Please consider of putting the validation data into a table. The senteces are very hard to read and follow.
All validation data have been included as Tables 1-3 and as Figure 3.
- Have you obtained Deming regression curves?
Deming regression curves could not be obtained since each blood sample was processed as a single aliquot by employing each of the three approaches presented. Instead, the results of Passing-Bablok regression analysis, which could be performed by processing each sample once, has been added to the manuscript (lines 291-295, 421-432, 497, 513-514, and Figure 6). The statistical descriptors of slopes and intercepts are provided in Table 4, also added as part of the revision.
- have you obtained Lin’s concordance correlation coefficients?
Lin’s concordance correlation coefficients are now shown in Figure 6. The methodology is described in lines 298-299, and the results are also presented in the main text (lines 432-439 and 497).
- Please include MRM chromatograms into supplementary data
MRM chromatograms of the analytes and the internal standards have been included in the Supplementary file. A reference to this has been added in line 110.
- It would be valuble to draw additional conclusion based on the obtained results as it would be valuble for readers, but also for clinical practice.
The Conclusions section has been updated. Additional conclusions have been drawn, with a special focus on the clinical implementation of various approaches to API and RIV assays (lines 527-528, 530, and 532-544).
